# The Interstitial Gland as a Source of Pro- or Anti-Senescent Cells during Chinchilla Rabbit Ovarian Aging

**DOI:** 10.3390/ijms25189906

**Published:** 2024-09-13

**Authors:** Verónica Díaz-Hernández, Alejandro Marmolejo-Valencia, César Montiel-De la Cruz, Gabriela Piñón-Zárate, Luis M. Montaño, Silvia Ivonne Mora-Herrera, Ivette Caldelas

**Affiliations:** 1Departamento de Embriología y Genética, Facultad de Medicina, Universidad Nacional Autónoma de México, Mexico City 04510, Mexico; 2Departamento de Biología Celular y Fisiología, Instituto de Investigaciones Biomédicas, Universidad Nacional Autónoma de México, Mexico City 04510, Mexico; alemarmol@biomedicas.unam.mx (A.M.-V.); caldelas@unam.mx (I.C.); 3Facultad de Ciencias, Universidad Nacional Autónoma de México, Mexico City 04510, Mexico; cmdlc@ciencias.unam.mx; 4Departamento de Biología Celular y Tisular, Facultad de Medicina, Universidad Nacional Autónoma de México, Mexico City 04510, Mexico; gabrielapinon@unam.mx; 5Departamento de Farmacología, Facultad de Medicina, Universidad Nacional Autónoma de México, Mexico City 04510, Mexico; lmmr@unam.mx; 6Unidad de Metabolómica y Proteómica, Instituto de Investigaciones Biomédicas, Universidad Nacional Autónoma de México, Mexico City 04510, Mexico; sivonnemor@iibiomedicas.unam.mx

**Keywords:** aging, senescent cells, ovary, rabbit, interstitial gland, lipophagy, aging ovary, p21, lipofuscin

## Abstract

The aging ovary in mammals leads to the reduced production of sex hormones and a deterioration in follicle quality. The interstitial gland originates from the hypertrophy of the theca cells of atretic follicles and represents an accumulative structure of the ovary that may contribute to its aging. Here, reproductive and mature rabbit ovaries are used to determine whether the interstitial gland plays a crucial role in ovarian aging. We demonstrate that, in the mature ovary, interstitial gland cells accumulate lipid droplets and show ultrastructural characteristics of lipophagy. Furthermore, they undergo modifications and present a foamy appearance, do not express the pan-leukocyte CD-45 marker, and express CYP11A1. These cells are the first to present an increase in lipofuscin accumulation. In foamy cells, the expression of p21 remains low, PCNA expression is maintained at mature ages, and their nuclei do not show positivity for H2AX. The interstitial gland shows a significant increase in lipofuscin accumulation compared with the ovaries of younger rabbits, but lipofuscin accumulation remains constant at mature ages. Surprisingly, no accumulation of cells with DNA damage is evident, and an increase in proliferative cells is observed at the age of 36 months. We suggest that the interstitial gland initially uses lipophagy to maintain steroidogenic homeostasis and prevent cellular senescence.

## 1. Introduction

In female mammals, the ovary is one of the first organs to age. A key feature of the aging ovary is a decrease in the follicular reserve, which reduces the production of sex hormones and deteriorates the quality of follicles, significantly impacting metabolic and reproductive health. Despite this, the mechanism underlying ovarian aging is not yet understood [1].

Senescence is a protective stress response mechanism that limits the replication of damaged or aged cells. Senescent cells can influence normal tissue homeostasis, aging, and disease. It has been proposed that senescence is a progressive step that may be associated with the heterogeneity of associated phenotypes in vivo. Their accumulation in vivo mediates adverse effects on tissue homeostasis [2].

The presence of cells with the senescence-associated secretory phenotype (SASP) in the ovary has been postulated. This phenotype is linked to an increased expression of cyclin-dependent kinase inhibitors, such as p16 and p21^Waf/CIP1^; the phosphorylation of the histone H2A variant (γH2AX) at the site of DNA damage; the accumulation of lipofuscin; and the secretion of cytokines and growth factors that promote a chronic inflammatory state.

In a recent study, Ansere and collaborators demonstrated that the presence of senescence-related markers preceded the decay of the follicular reserve [3]. Furthermore, Maruyama and colleagues reported senescence-associated β-galactosidase staining of CDKN2A (p16) in aged ovarian stroma cells of aged mouse ovaries. They also found that some transcripts of SASP factor expression were elevated in aged ovaries [4]. Furthermore, Hense and coworkers showed the accumulation of more senescent cells in obese mice. However, it was not possible in all cases to identify the senescent cell type [5,6]. Another marker of senescence is lipofuscin accumulation, which has also been reported to increase with age in mouse ovaries [3,5,7]. However, the specific cell type exhibiting higher accumulation has not yet been identified [6].

Moreover, previous studies reported that a population of multinucleated macrophage giant cells, characterized by a foamy appearance and containing vacuoles and inclusion bodies, is present in aged ovarian tissue but absent in the ovaries of young mice [8,9]. The authors proposed that these macrophage foamy cells may mediate ovarian aging inflammation. However, the effect of macrophage accumulation in the aged ovary remains undetermined. 

Although the reports mentioned above suggest the presence of senescent cells in the ovary, which may promote an inflammatory state leading to aging, the cell type that acquires senescent characteristics has not yet been determined. We are even further away from understanding whether possible ovarian inflammation is a cause or a consequence leading to ovarian aging. 

To understand ovarian aging, it is important to note that ovarian tissue has a dynamic microenvironment that undergoes constant structural changes with each menstrual cycle. One significant histological change in ovarian development is the formation of the interstitial gland. In mammalian ovaries, only close to 1% of follicles ovulate, whereas 99% undergo atresia [10]. The destiny of the theca interna of atretic follicles is to produce steroidogenic tissue, known as the secondary interstitial gland [11]. Unlike the luteal cells that result from one ovulatory follicle, the cells of the interstitial gland originate from the hypertrophy of the theca cells of atretic follicles and represent a permanent and accumulative feature of the ovary, which suggests that they may contribute to ovarian aging. The secondary interstitial gland, known succinctly as the interstitial gland, is a tissue that has only been somewhat investigated but is abundant in the ovaries of various species: pig, guinea pig, marmoset, human, rabbit, hare, bat [11], gerbil [12], and rat [13]. In all these species, the cells of the interstitial gland test positive for Sudan Black B staining, as documented by Guraya in 1978 [11]. Recently, we showed that, in rabbits, the hypertrophic interstitial gland became the most abundant tissue in the aging ovary [14]. 

In 1953, Claesson and Hillarp, as well as Guraya and Greenwald in 1964, reported that the interstitial gland in rabbits stores cholesterol, phospholipids, acetal phospholipids, and residual fatty acids, as well as triglycerides contained in lipid droplets (LDs) [15,16]. Recently, LDs have been recognized as independent organelles that regulate the dynamics of lipid uptake and metabolism; form a link between cells and the environment; and provide substrates for energy, membrane synthesis, and production molecules associated with lipids and steroid hormones [17]. Many signs of aging are linked to significant changes in LD functions, such as a reduced ability to use lipids as energy sources or an increased abnormal lipid buildup. Problems with LD regulation can harm aging processes, leading to degenerated proteasomal and mitochondrial activities and the induction of cellular senescence [18]. It has recently been described that lipophagy is the process by which different cell types, such as adipocytes, foamy macrophages, enterocytes, and neurons, regulate lipid distribution and protect against lipid accumulation [17] and, in the case of luteinized granulosa cells, regulate steroidogenesis [19]. 

Lipophagy is a type of autophagy defined as the auto-degradation of intracellular LDs [20]. Although the effects of autophagy on organelle recycling have been known since the 1960s, the contribution of autophagy to LD degradation has only recently been reported [20]. The contribution of autophagy to LD degradation was first demonstrated in hepatocytes, which is where the term lipophagy emerged from [21]. Although the role of lipophagy in the process of cellular aging is still incipient, it has been stated that the malfunctioning of the different types of autophagy with age affects other determinants of aging such as cellular senescence, DNA alterations, and cellular communication dysfunction, placing autophagy at the center of cellular aging processes [22].

Considering that the interstitial gland is a steroidogenic tissue that accumulates with age, characterized by the presence of LDs and lipofuscin, this study aimed to determine whether the interstitial gland plays a crucial role in ovarian aging as a possible source of senescent cells. 

## 2. Results

### 2.1. The Interstitial Gland Gradually Replaces the Follicular Component That Occupies the Cortical Ovary

To demonstrate the histological changes that occur in the ovary, we used semi-thin sections of approximately 1 µm thickness stained with toluidine blue. 

We used a 4-month-old female rabbit as a reproductive model. At this stage, we observed atresia in several antral follicles, indicating that the formation of the secondary interstitial gland had initiated (Figure 1A). Moreover, many preantral and antral follicles were observed. An emerging feature observed in the interstitial gland cells, from the moment of their formation in the ovaries of 4-month-old rabbits to mature rabbits, was the presence of lipid droplets (Figure 2). In young ovaries, an ultrastructural study using transmission electronic microscopy showed that these cells contained organelles typical of steroidogenesis: round and elongated mitochondria with tubular cristae, a smooth endoplasmic reticulum, and lipid droplets easily identifiable as round, light-density structures not limited by a bilayer lipid membrane (Figure 2B,C). Heterochromatin was located in clusters at the periphery of the nuclear membrane. Most of the nucleus contained euchromatin, indicating that these cells were transcriptionally active. 

In a previous study, we used 16–36-month-old female rabbits as an aging model. We reported a significant decrease in the follicular reserve at this stage [14]. In mature ovaries, the interstitial gland was extensively developed in the form of a lobule conforming to various sizes, which occupied the cortical region, but its extension was delimited by the tunica albuginea (Figure 1B–D). In 18-month-old rabbit ovaries, we observed developed antral follicles and abundant oocyte remnants between the lobules of the interstitial gland (Figure 1B). By 24 months, evidently, the lobule in the rabbit ovary became compacted, and new hypertrophied cells with a polyhedral appearance were incorporated in the lobules of the interstitial gland (Figure 1C, red arrow). By 36 months, the rabbit ovaries showed a more compact lobule. In the representative image, it is still possible to observe an antral follicle, which is surrounded by the interstitial gland (Figure 1C). 

Interestingly, interstitial gland foamy cells were commonly found in the ovaries of the mature rabbits, typically organized in a septal pattern between the interstitial gland lobules. They were generally characterized by round nuclei and a poorly stained foamy cytoplasm with an irregular form (Figure 3A,B). Similar foamy cells have been reported in aged mouse ovaries, and they have been characterized as macrophages [8,9]. However, in this case, an ultrastructural study showed that the foamy cells conserved round mitochondria with tubular cristae and a smooth endoplasmic reticulum from which small lipid droplets and abundant lipid droplets seemed to emerge (Figure 3C,D). We thus believe that the cells of the interstitial gland transformed into cells resembling foamy cells due to the accumulation of lipid droplet clusters. The size of the lipid droplet clusters varied considerably from one cell to another (Figure 3C–F).

### 2.2. The Ovarian Interstitial Gland Foamy Cells (OIGFCs) Show Ultrastructural Characteristics of Lipophagy

Lipophagy is a type of autophagy that involves the degradation of lipid droplets within cells. It plays a role in lipid distribution and protection against lipid accumulation [18]. This process has been observed in various cell types, including adipocytes, macrophages, foamy cells, enterocytes, hepatocytes, and neurons, as well as in cells of the human ovary and testis, where it is necessary for steroid synthesis [17,19]. In this context, cells now known as ovarian interstitial gland foamy cells (OIGFCs) manifest the lipophagy process. Lysosome arm-like extensions could be seen engulfing lipid droplets (Figure 4A) and multivesicular bodies (mvb) (Figure 4B). In others, foamy cells presented smaller and darker mitochondria (Figure 4C), suggesting a defect in steroidogenesis. Lipid-containing double-membrane vesicles, similar to autolipophagosomes (aphs) (Figure 4D,E), were apparent. However, no alterations were observed at the level of nuclear morphology (Figure 4A,B).

### 2.3. The OIGFCs Do Not Express the Pan-Leukocyte Marker CD-45 but Do Express CYP11A1

To confirm that the OIGFCs were not derived from lymphoid cells, such as macrophages, we utilized an antibody targeting CD-45, a glycoprotein expressed in all hematopoietic cells, except for mature erythrocytes and platelets. This is considered a pan-leukocyte marker [23]. As a positive control tissue, we used 18-month-old rabbit liver slices. An intense stain was observed in Kupffer cells (macrophages of the liver) (Figure 5A). In the ovary, we observed some CD-45-positive cells, similar to macrophages close to the medullar region (Figure 5B,C). However, the interstitial glandular foamy cells were negative for CD-45 (Figure 5D). Furthermore, as the cells of the interstitial gland exhibited steroidogenic activity, we sought to immunolocalize the expression of Cytochrome P450 family 11 subfamily A member 1 (CYP11A1), which catalyzes the first and rate-limiting step of the steroid hormones in the mitochondrial inner membrane, converting cholesterol into pregnenolone [24]. There was a notably abundant expression of CYP11A1 in the interstitial gland (Figure 5E,F). The intensity varied in the interstitial foamy cells, indicating a potential disturbance in steroidogenesis, as was apparent in the mitochondrial ultrastructure. However, this expression was still evident compared with other cells in the stromal compartment and the negative control (Figure 5F). 

### 2.4. The Interstitial Gland as a Source of Anti-Senescent Cells 

In one of our previous studies, we observed that the interstitial gland accumulates lipofuscin, which is considered “the aging pigment”. In order to determine the presence of aging markers, we used Sudan Black B (SBB) staining to identify lipofuscin accumulation in young and mature rabbit ovaries (Figure 6). A quantitative analysis revealed that, at 18 months old, higher lipofuscin accumulation was observed in the surface epithelium and the interstitial gland of the mature rabbit ovaries when compared with other structures. Additionally, the most intense SBB staining was observed in OIGFCs (Table 1). At 24 and 36 months, the OIGFCs showed the greatest lipofuscin accumulation compared with other structures. Furthermore, lipofuscin accumulation remained constant with age in the interstitial gland and OIGFCs (Table 1). Likewise, we immunolocalized another marker of cellular senescence, p21^Waf/CIP1^, and we complemented our analysis with the detection of proliferating cell nuclear antigen (PCNA), expressed only in proliferating cells. 

In the ovaries of the reproductive and mature 18- and 24-month-old rabbits, PCNA expression was evident in the surface epithelium (Figure 7H) and growth follicles (Figure 7A,D,G). In the interstitial gland, the cells close to the surface epithelium expressed PCNA (Figure 7B,C,E,H) and did not show significant differences between the ages studied (Figure 8A). As these cells incorporated and accumulated in the section closest to the medullar region, the percentage of PCNA-positive cells decreased significantly at 18 and 24 months old (Figure 7F,I and Figure 8A; ** *p* < 0.01, ** *p* < 0.001). Additionally, there were significant differences in the percentage of PCNA-positive cells localized close to the surface epithelium in a comparison with the medullar region at 18 and 24 months old (Figure 8A * *p* < 0.05 and *** *p* < 0.001). However, in the ovaries of the 36-month-old rabbits, most of the interstitial gland cells were PCNA-positive regardless of whether they were localized near the epithelium or the medullary region (Figure 7J–L and Figure 8A). The expression of p21^Waf/CIP1^ in the interstitial gland decreased significantly in the comparison between the 4-month-old and 36-month-old rabbits (Figure 7C’; Figure 8B, * *p* < 0.05) and between the 24-month-old and 36-month-old rabbits (7F′,I′,L′); Figure 8B, ** *p* < 0.01).

Moreover, in the young and mature rabbit ovaries, γH2AX (DNA damage marker) was found in all oocytes (due to genetic recombination) (Figure 9A), in the follicular cells of the atretic follicles (Figure 9B,D,G,J) and in the interstitial cells of the antral atretic follicles during their final stage (Figure 9C,E,K). In the mature rabbit ovaries, γH2AX was evident in some interstitial gland cells, especially in the most recently formed lobules and cells close to atretic follicles. A negative expression of γH2AX is evident in the old interstitial gland (Figure 9F,L), the percentage of γH2AX-positive cells in the interstitial gland significantly decreased at 24- and 36-month- old compared with at 4 months old (Figure 8C, * *p* < 0.05; ** *p* < 0.01; Figure 9H,I).

Interestingly, 100% of the OIGFCs were positive for PCNA expression (Figure 10A), and 100% did not show γH2AX labeling (Figure 10C)) and had weak p21^Waf/CIP1^ expression compared with the interstitial gland (** *p* < 0.01; *** *p* < 0.001, * *p* < 0.05, respectively) (Figure 8B; Figure 10B). 

## 3. Discussion

It has been hypothesized that the accumulation of senescent cells contributes to ovarian aging. Some studies have searched for markers of cellular senescence in the ovary, mainly by examining the expression of transcripts of various factors involved in senescence [3,4,5]. However, these studies have focused on overall tissue analyses, with little attention paid to morphological aspects, except for follicle counts or low-resolution immunofluorescence. As a result, the ovary’s specific cell type and senescent phenotype have not been described. The closest approximation to date has been the detection of senescent factors in the ovarian stroma. Considering the histological changes as the ovary ages, our work aimed to determine whether the ovarian interstitial gland could be a potential source of senescent cells. In the ovary, there are different types of interstitial tissue, such as the corpus luteum, fetal primary interstitial gland (which appears during the fetal or neonatal and postnatal periods), and theca cells. However, this work focused on the secondary interstitial gland. The somatic compartments of antral follicles contain two highly steroidogenic cell types: granulosa cells and theca interna cells. During atresia, which is the fate of 99% of ovarian follicles, granulosa cells undergo apoptosis, whereas theca interna cells undergo hypertrophy and reorganize to form the secondary interstitial gland, retaining steroidogenic capacity [14].

Here, we found that the interstitial gland gradually replaces the follicular component; it occupies most of the ovarian cortex, except for the tunica albuginea. Unlike the luteal cells resulting from one ovulatory follicle, the secondary interstitial gland is a permanent feature in the ovary, whose role during ovarian aging requires further investigation. 

In 1964, Mossman and collaborators stated that the term “interstitial cells” is inaccurate when applied to the ovary because it ignores its existence as a permanent and functional gland-like tissue in the ovary [25]. This has also resulted in it being omitted from the records of several species and even from recent works that require the identification of senescent cells.

Here, we demonstrated that, in the mature ovary, the interstitial gland cells accumulate lipid droplets and show ultrastructural characteristics of lipophagy. Due to their steroidogenic activity, it appears likely that these cells undergo the processes of lipophagy and lipolysis in order to maintain the homeostasis of the glandular tissue and avoid lipotoxicity. Recently, Esmaeilian and colleagues (2023) demonstrated that lipophagy regulates the degradation of lipid droplets to release the free cholesterol required for steroidogenesis in the human ovary and testis, meaning that this process is defective in women with luteal phase defects [19,26]. Therefore, further studies on the role of lipophagy are crucial to understanding the mechanism that causes it. Likewise, it is important to understand this process in other cell types that present steroidogenesis, such as aging theca interna cells and granulosa cells. Autophagy is a process that involves the recycling of proteins and organelles, mediated by lysosomal degradation, and it is considered essential for cell survival.

The different types of autophagy, including lipophagy and senescence, share several stimuli, including telomere shortening, DNA damage, and oxidative stress, suggesting that they have an intimate relationship. Recently, evidence corroborated an anti-senescence role played by autophagy that acts in the homeostatic pathway and relieves the burden of senescence-inducing stressors; the pro-senescence role occurs under pathological conditions. This emphasizes the discrimination between the role of basal autophagy in the normal aging process as opposed to in a pathological process [27]. Considering that lipophagy is a special type of autophagy, we suggest that the interstitial gland initially uses lipophagy to maintain steroidogenic homeostasis and prevent cellular senescence. We observed that OIGFCs reduce the steroidogenic expression of CYP11A1 (an enzyme that catalyzes the conversion of cholesterol to pregnenolone). Their nucleus did not show the compaction of chromatin and expressed PCNA, as well as showing a decreased expression of p21^Waf/CIP1^. Moreover, older interstitial gland cells did not show DNA damage. The homeostasis-mediated process of lipophagy helps to explain why the accumulation of lipofuscin remains constant in mature ovaries at different ages. Additionally, the expression of PCNA serves as a complementary mechanism to reduce its accumulation. However, when this process is disrupted, possibly due to aging, it may lead to senescence. Our ability to analyze long-lived rabbits in our study model is compromised by the time for which they must be maintained. 

In addition, the upregulation of lipolysis may damage cells by increasing the pool of free fatty acids. This provokes progressive lipotoxicity, the accumulation of reactive oxygen species, organellar stress, and death. The intracellular accumulation of lipid droplets is a hallmark of diseases: obesity, steatosis, diabetes, myopathies, and arteriosclerosis [18]. 

In previous studies multinucleated giant cells with a foamy aspect were identified as macrophages because they stained positive for pan-marker macrophages. However, their expression was variable in positive cells, being reduced or absent [8], indicating fusogenic activity between macrophages and other lineages [9]. Although the structure of the interstitial gland in the mouse ovary is less evident than that in the rabbit ovary, in this case, macrophages may be associated with the interstitial gland, but whether they are involved in lipophagy activity remains to be determined.

## 4. Materials and Methods

This research was carried out with approval from the local Ethics Committee of the Facultad de Medicina, UNAM. Animals were sacrificed via an intravenous pentobarbital overdose (60 mg/Kg) (Pisa Agropecuaria, CDMX, Guadalajara, Jalisco, Mexico), applied through the marginal ear vein. All procedures were carried out in compliance with the National Research Council’s “Guide for the Care and Use of Laboratory Animals”.

We used 20 female rabbits from the European chinchilla breed (Oryctolagus cuniculus). Ovaries from females of different ages were collected and classified into 2 groups: reproductive non-mated estrous, 4 months of age (N = 5); and mature, up to 18, 24, and 36 months of age (N = 5 per age). The mature females were multiparous and showed a decrease in fertility, after which they rejected copulation. 

The ovaries were quickly removed and fixed as follows: The left ovary was cut in half along the transverse plane and fixed in cold 4% paraformaldehyde (Sigma-Aldrich, St. Louis, MO, USA) overnight. Then, it was dehydrated, cleared in xylene, and embedded in paraffin. A rotary microtome was used to section tissues at 7 µm, and alternate slices were mounted on glass slides for histology, Sudan Black B (SBB) staining, and immunohistochemistry. 

The right ovary was removed, and a thin slice (approximately 1.5 mm) from the central zone was immediately collocated in Karnovsky fixative; here, we cut the slice in half to improve fixation during a 24 h period.

### 4.1. High-Resolution and Transmission Electron Microscopy

The slice fixed in the Karnovsky fixative was subsequently postfixed with 1% OsO_4_ in Zetterqvis’s buffer for 2 h at 4 °C, dehydrated with an ascending series of ethanol, and then infiltrated and embedded in Epon 812. 

Semi-thin sections of 1 µm were cut using an ultramicrotome, stained with toluidine blue for orientation, and studied using light microscopy. Ultrathin sections were placed on a copper square mesh grid (Cat. 7552C3, Pelco, Fresno, CA, USA), stained with uranyl acetate, and examined using a JEOL electron microscope 1010.

### 4.2. Lipofuscin Detection and Analysis

Lipofuscin detection was performed with SBB staining. Briefly, paraffin was removed and rehydrated in a descending ethanol series. The slides were stained with SBB 0.7% (Cat. No. 199664) (Sigma-Aldrich, St, Louis, MO, USA) in 70% ethanol for 2 min. Subsequently, the sections were briefly rinsed with 50% ethanol and distilled water. Finally, they were mounted using an aqueous mounting medium (F4680, Sigma-Aldrich, St Louis, MO, USA). The sections were examined using a Nikon microscope, model Eclipse Ni-U (Nikon, Tokyo, Japan), attached to a digital camera at 10× magnification and using NIS-Elements AR 4.60.00 software. For qualitative analyses, lipofuscin quantification in images of ovarian structures was performed using Image J/Fiji software. The optical density (OD) was determined for each sample with Image J/Fiji software immunostaining for quantification. This was measured as a mean gray value in the region of interest. The optical density was calculated using the formula OD = log (max intensity/mean intensity), where the max intensity is 255 for 8-bit images.

### 4.3. Immunohistochemistry 

We rehydrated the paraffin sections and performed heat-induced epitope retrieval in a decloaker chamber for 15 min at 110 °C in 1X Diva (Biocare Medical, Pike Lane, CA, USA) or Tris-buffer EDTA pH 10 (see Table 2). Then, the slides were placed on a sequenza immunostaining rack and incubated with a blocking solution (10% horse serum, 2% serum albumin bovine in 0.5% triton/TBS) for 1 h at 32 °C. Endogenous peroxidases and biotin were quenched with Bloxall (SP-6000, Vector Laboratories, Burlingame, CA, USA) for 20 min and a Streptavidin/Biotin-Blocking Kit (Vector laboratories, Burlingame, CA) for 15 min, respectively. They were then incubated with primary antibodies (Table 2) diluted in Da Vinci Green diluent at 4 °C overnight. Each immunostaining run included negative controls, for which primary antibodies were replaced with the incubated Da Vinci Green Diluent (Biocare Medical, Pike Lane, CA, USA) without primary antibodies. Immunostaining with the same primary antibody was carried out simultaneously for all ages studied. The tissue sections were washed with TBS three times for 5 min. The corresponding secondary antibodies were incubated for 20 min at room temperature (Table 2). The slides were incubated with a Betazoid DAB Chromogen Kit (BDB2004, Biocare, Pike Lane, CA, USA) to reveal the color of the enzyme–substrate antibody staining. Finally, they were treated with acetate buffer pH 5 for 10 min and counterstained with hematoxylin for 2 min. The slides were dehydrated in a gradual series of alcohol, transferred to xylene for 5 min, and mounted with Sub-X Mounting Medium. The sections were examined using a Nikon microscope, model Eclipse Ni-U (Nikon, Tokyo, Japan), attached to a digital camera at 40× magnification and using NIS-Elements AR 4.60.00 software. The percentage of PCNA- and γH2AX-positive cells was evaluated (total number of interstitial gland cells/number of positive cells × 100) in 4 ovary sections of each animal in each age. The optical density (OD) was determined for each sample with Image J/Fiji software immunostaining for quantification. This was measured as the mean gray value in the interstitial gland region. The optical density was calculated using the formula OD = log (max intensity/mean intensity), where the max intensity is 255 for 8-bit images. Antibody validation was performed using Western blotting. We used 0.3 g of the 30-month-old rabbit ovary snap frozen in liquid nitrogen; the sample was homogenized in RIPA lysis buffer (sc-24948, Santa Cruz Biotechnology, Dallas, TX, USA) with a polytron PI 1200E (Kinematica AG, Malters, Switzerland) for a few seconds until the tissue was completely dissolved. The lysate was cleared via spinning for 20 min at 13,000 rpm and 4° C. The protein was quantified using a Pierce 660 nm protein assay reagent (1861426 Thermo Scientific, Rockford, IL, USA). We used 100 µg of protein in a precast gel Bolt 4-12% Bis-Tris Plus (NW04120BOX, Invitrogen, Rockford, IL, USA). The proteins were transferred to mini-sized nitrocellulose transfer stacks (PB3210, Invitrogen, Rockford, IL, USA) using a power blotter system (PB0012, Invitrogen, Rockford, IL, USA). Next, these were incubated in 4% blocking solution (Blotto, nonfat dry milk sc-2325, Santa Cruz Biotechnology, Dallas, TX, USA) for one hour and then incubated with primary antibodies (Table 2) diluted in blocking solution. The membranes were washed with TBS–Tween (0.1%) and then incubated with ECL Prime Western Blotting Detection Reagents (RN2232, Amersham, Rockford, IL, USA). Western blots were imaged digitally with an Alliance 4.7-UVITEC imaging system and processed with Alliance ID 4.7 software. The results of the antibody validation are available in Appendix A.

### 4.4. Statistical Analysis 

Data are expressed as the mean ± the standard error of the mean (SEM). Statistical differences between groups were determined using a one-way ANOVA, followed by a Tukey–Kramer Multiple Comparisons test using GraphPad InStat3 Software 3.05. Statistical differences between the two groups were established by applying a paired Student’s *t*-test. Statistical significance was set to & and * *p* < 0.05, && and ** *p* < 0.01, and *** *p* < 0.001.

## 5. Conclusions

We conclude that the interstitial gland initially uses lipophagy to prevent the senescence of cells. However, their dysregulation, which may be age-dependent, may also promote senescence. Clarifying the roles of lipid droplets and lipophagy in the ovary is important for developing an understanding of the mechanism used to overcome the effects of age and to compare these responses with those to external stress factors such as fatty diets and pollution. This work has opened new avenues for future research and interpretation. 

## Figures and Tables

**Figure 1 ijms-25-09906-f001:**
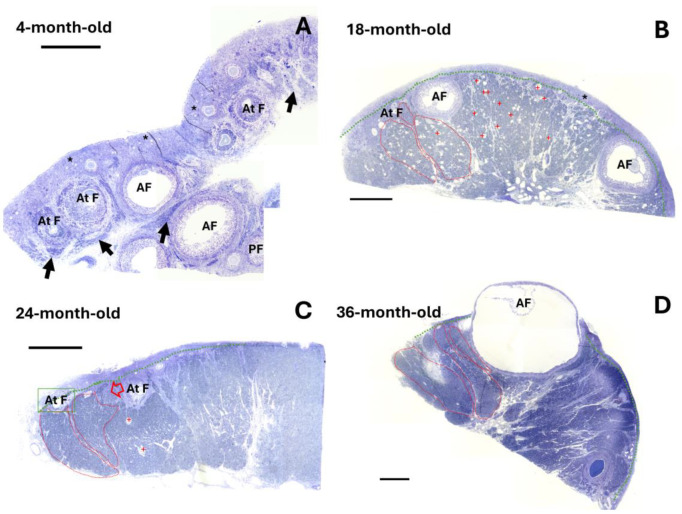
Representative semi-thin sections of ovaries from rabbits in reproductive and mature stages. (**A**) Image of a semi-thin section of an ovary from a 4-month-old rabbit, displaying numerous preantral follicles (PF), antral follicles (AF), and atretic follicles (At F) and clusters of primordial follicles (*); interstitial gland undergoing formation (black arrows). (**B**–**D**) Images of semi-thin sections of ovaries from 18-, 24-, and 36-month-old rabbits, respectively, showing extensively developed interstitial gland-forming lobules, delineated by red dotted lines. The interstitial gland replaces the follicle component and begins to take up space in the cortical ovary. The green line delimits the tunica albuginea. Antral atretic follicles are also visible, as well as oocyte remnants (red crosses). Scale 500 µm.

**Figure 2 ijms-25-09906-f002:**
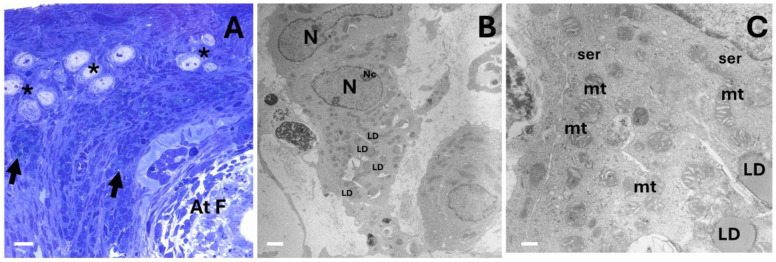
Representative light and transmission electron microscopy images showing the ovary of a 4-month-old rabbit. (**A**) Semi-thin section displaying a cluster of primordial follicles (*), the atretic follicle (At F), and many cells of the interstitial gland with the characteristic presence of lipid droplets (black arrows). (**B**) Electron micrograph showing the interstitial gland cell with its nucleus (N) and large nucleolus (Nc) and lipid droplets (LD). (**C**) Close-up of part of the interstitial gland, showing round and elongated mitochondria with tubular cristae (mt), a smooth endoplasmic reticulum (ser), and easily identifiable round, light-density lipid droplets (LD). Scale: (**A**) 20 µm, (**B**) 2 µm, (**C**) 50 nm.

**Figure 3 ijms-25-09906-f003:**
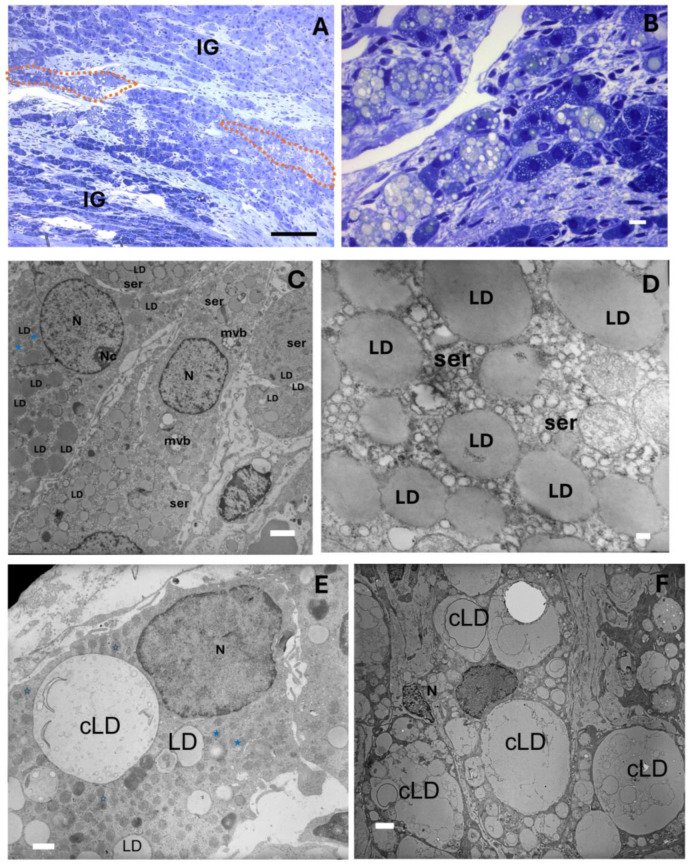
Representative light and transmission electron microscopy images of a 24-month-old rabbit ovary. (**A**) Semi-thin section showing lobules in the interstitial gland (IG), where cells with a foamy appearance and an irregular form are evident, delineated by red dotted lines. (**B**) Foamy cells of the interstitial gland at a greater magnification. (**C**) An electron micrograph of interstitial gland cells displaying round mitochondria (blue stars) with tubular cristae and a smooth endoplasmic reticulum (ser) from which small lipid droplets and abundant lipid droplets can be seen emerging (see close-up in (**D**)). (**E**) Electron micrograph of interstitial gland cells displaying clusters of different-sized lipid droplets; however, these coexist with typical steroidogenic organelles and the smaller and darker mitochondria (blue stars) described in the previous micrograph. (**F**) Electron micrograph of different interstitial gland foamy cells; the size of the lipid droplet clusters (cLD) varies considerably. LD = lipid droplet, ser = smooth endoplasmic reticulum, N = nucleus, mvb = multivesicular bodies. Scale bar: (**A**) 100 µm, (**B**) 10 µm, (**C**) 2 µm, (**D**) 200 nm, (**E**) 1 µm, (**F**) 2 µm.

**Figure 4 ijms-25-09906-f004:**
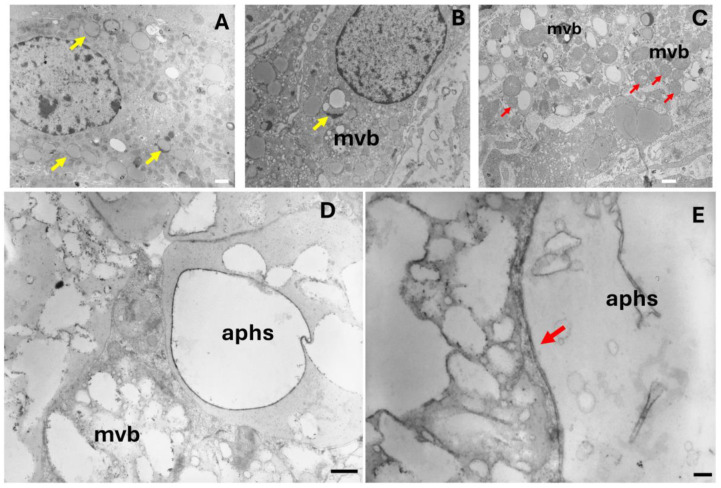
Transmission electron microscopy of 24- and 36-month-old rabbit ovaries. (**A**–**C**) Electron micrograph of interstitial gland foamy cells displaying lysosome arm-like extensions (yellow arrows), multivesicular bodies (mvb), and darker mitochondria (red arrows). (**D**) Image showing lipid-containing double-membrane vesicles similar to autolipophagosomes (aphs). (**E**) Close-up view of part of lipid-containing double-membrane vesicles (red arrow). Scale bar: (**A**–**C**) 1 µm, (**D**) 500 nm, (**E**) 200 nm.

**Figure 5 ijms-25-09906-f005:**
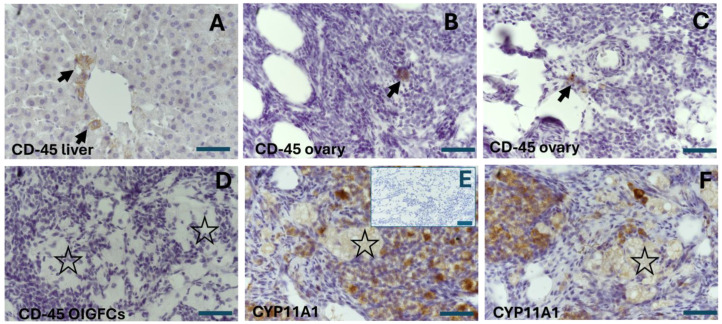
Representative immune detection images of CD-45, a pan-leukocyte marker, and CYP11A1 enzyme. (**A**) A liver section was employed as a positive control tissue for CD-45 detection, and an intense stain was observed in Kupffer cells (dark arrows). (**B**,**C**) In the ovary of a 24-month-old rabbit, some CD-45-positive cells, similar to macrophages, are visible close to the medullar region (black arrows). (**D**) Ovarian interstitial glandular foamy cells (OIGFCs) were negative for CD-45 (dark stars). (**E**,**F**) Immunohistochemistry revealed an abundant expression of CYP11A1 in the interstitial gland in OIGFCs (dark stars), with insert showing a negative control without anti-CYP11A1, using anti-goat secondary antibody. The intensity varied; however, this expression was still evident when compared with other cells in the stromal compartment and the negative control. Scale bar: 50 µm.

**Figure 6 ijms-25-09906-f006:**
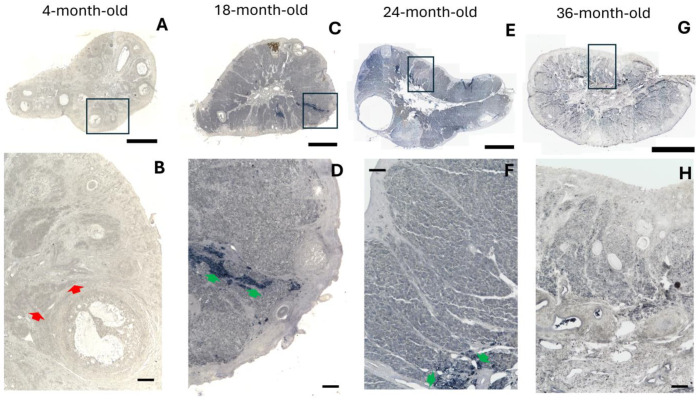
Representative photomicrographs used to detect lipofuscin through Sudan Black B (SBB) staining of reproductive and mature ovaries of rabbits. (**A**) A weak stain is visible in the ovary of the reproductive rabbit. (**B**) The interstitial gland is marked with red arrows. (**C**,**E**,**G**) The interstitial gland of the mature rabbit ovaries shows lipofuscin accumulation, and OIGFCs show greater accumulation. (**D**,**F**,**H**) Amplification of the area marked in the top panel; note the foamy appearance of the OIGFCs (green arrows). Scale bar: (**A**,**C**,**E**,**G**) 300 µm; (**B**,**D**,**F**,H) 100 µm.

**Figure 7 ijms-25-09906-f007:**
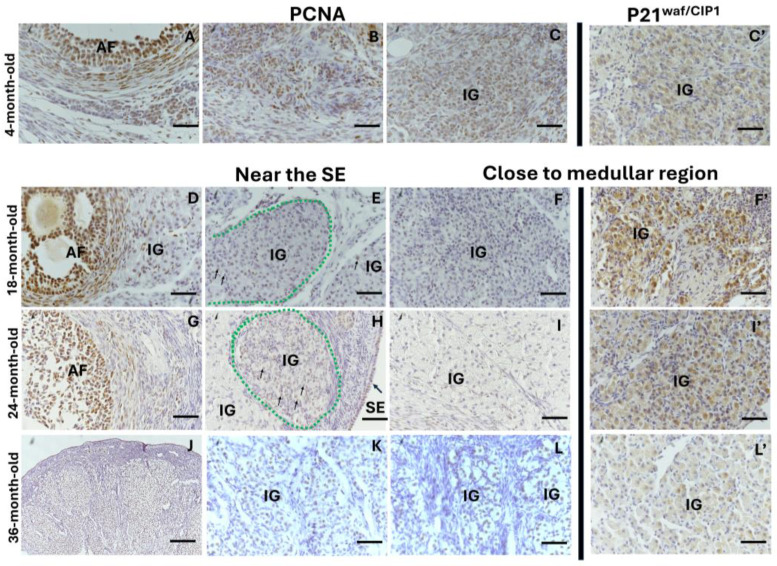
Immunohistochemical detection of proliferating cell nuclear antigen (PCNA) and a marker of cellular senescence, p21^Waf/CIP1^, in the ovaries of reproductive and mature rabbits. Labels on the left side indicate the age of the rabbit ovaries. Labels at the top indicate the protein identified, and the second set of labels indicates the region (near the surface epithelium (SE) or close to the medullar region). (**A**–**C**) Nuclear staining for PCNA in granulosa and theca cells of the antral follicle (AF), atretic follicle (At F), and recently formed interstitial gland (IG). (**D**–**H**) PCNA-positive antral follicles within a region (delineated by a green dotted line) that is positive for PCNA (thin arrows) near the surface epithelium (SE) and the interstitial gland that is negative for PCNA staining. (**I**) IG negative for PCNA. (**J**) Lower magnification reveals an interstitial gland area with cells positive for PCNA, regardless of their location. (**K**,**L**) These areas at greater magnification. (**C′**,**F′**,**I′**,**L′**) Immune-detected p21 in interstitial gland cells, close to the medullar region. Scale bar: (**A**–**I**,**K**,**L**) 50 µm; (**J**) 100 µm; (**C′**,**F′**,**I′**,**L′**) 50 µm.

**Figure 8 ijms-25-09906-f008:**
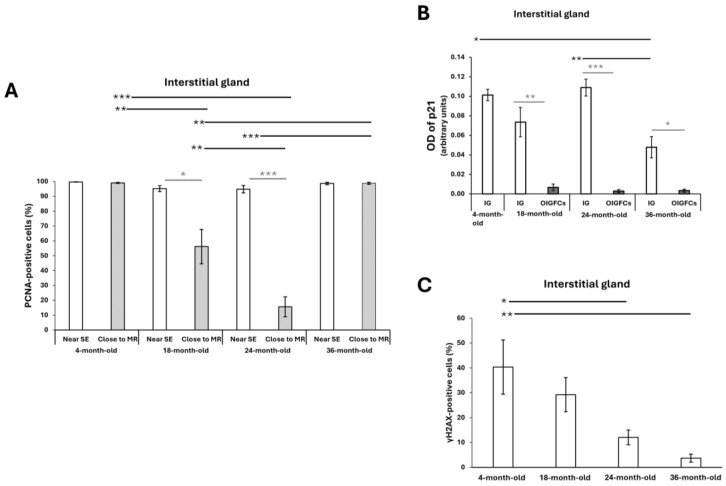
Analysis of (**A**) percentage of cells positive for PCNA, (**B**) densitometric quantification performed on immunohistochemical localization of p21 in the interstitial gland and OIGFCs (ovarian interstitial gland foamy cells), and (**C**) percentage of cells positive for γH2AX in ovarian sections from reproductive and mature chinchilla rabbits. Data are expressed as mean ± SEM; * *p* < 0.05; ** *p* < 0.01; *** *p* < 0.001. Near SE: near to surface epithelium; close to MR: close to medullar region.

**Figure 9 ijms-25-09906-f009:**
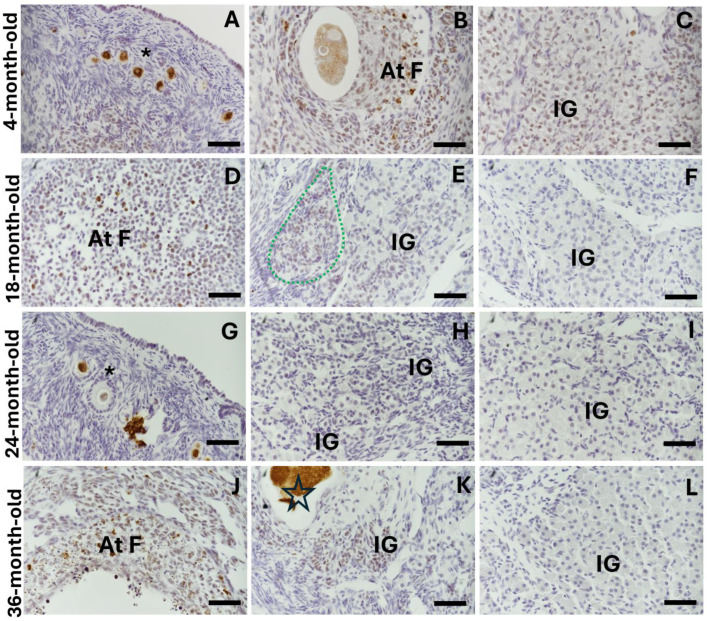
Photographs of immunohistochemical staining of γH2AX (damaged DNA) in the ovaries of reproductive and mature rabbits. (**A**–**C**) Nuclear staining for γH2AX is visible in oocytes (*) (this is a marker of recombinant meiotic products), atretic follicle (At F), and recently formed interstitial gland (IG). (**D**–**F**) γH2AX detection in atretic follicles (dotted green line in (**E**)) and negative expression of γH2AX in the old interstitial gland. (**G**) γH2AX in oocytes of primordial and primary follicles. (**H**,**I**) Interstitial gland negative for γH2AX. (**J**) Atretic follicle cells positive for γH2AX. (**K**) An oocyte remnant (black star) and somatic cells of the atretic follicle positive for γH2AX. (**L**) Interstitial gland cells negative for γH2AX. Scale bar: 50 µm.

**Figure 10 ijms-25-09906-f010:**
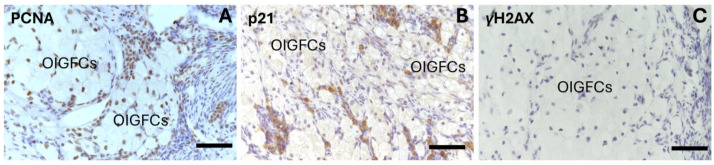
Immunohistochemical staining of PCNA, p21^Waf/CIP1^, and γH2AX in ovarian interstitial gland foamy cells (OIGFCs) of 24-month-old rabbits. (**A**) Section shows PCNA nuclear brown staining in the nuclei of OIGFCs. (**B**) Negative p21^Waf/CIP1^ expression in OIGFCs and positive expression in nearby cells. (**C**) OIGFCs do not indicate any damage to DNA (negative for γH2AX). Scale bar: 50 µm.

**Table 1 ijms-25-09906-t001:** Comparison of the optical density (OD) of SBB-stained sections from the different ovarian structures of reproductive and mature rabbits.

	Mean OD SBB ± SEM.			
Structure	4-Month-Old	18-Month-Old	24-Month-Old	36-Month-Old
(a) Surface epithelium	0.059 ± 0.009	0.173 ± 0.023 ^&&^ h***	0.144 ± 0.025 ^&^ h***	0.1599 ± 0.005 ^&&^ h***
(b) Primordial follicles	0.044 ± 0.006	0.086 ± 0.009 ^&^ a**, g**, h***	0.103 ± 0.020 ^&^ h***	0.1058 ± 0.008 ^&^ h***
(c) Primary follicles	0.045 ± 0.008	0.087 ± 0.010 a**, g**, h***	0.101 ± 0.028 h***	0.0980 ± 0.008 h***
(d) Secondary follicles	0.045 ± 0.007	0.099 ± 0.012 ^&^ a*, g**	0.113 ± 0.019 ^&&^ h***	0.110 ± 0.009 ^&&^ h***
(e) Atretic follicles	0.048 ± 0.010	0.082 ± 0.008 a**, g**	0.114 ± 0.15 ^&&^ h***	0.0105 ± 0.009 ^&^ h***
(f) Oocytes	0.050 ± 0.010	0.085 ± 0.009 a**, g**	0.098 ± 0.020 h***	0.100 ± 0.0003 ^&^ h***
(g) Interstitial gland	0.061 ± 0.010	0.179 ± 0.016 ^&&^ h***	0.175 ± 0.054 ^&&^ h***	0.173 ± 0.023 ^&&^ h***
(h) OIGFCs	-	0.378 ± 0.004 ^&&^	0.398 ± 0.039 ^&&^	0.396 ± 0.045 ^&&^

Statistical results of the OD of SBB staining of ovarian structures: a* (*p* < 0.05); a** (*p* < 0.01) vs. surface epithelium, g** (*p* < 0.01) vs. interstitial gland, h*** (*p* < 0.01) vs. ovarian interstitial gland foamy cells (OIGFCs). Statistical results of the OD of SBB staining across ages: ^&^ (*p* < 0.05), ^&&^ (*p* < 0.01) as compared with the corresponding ovarian structure at 4-month-old. SEM= standard error of the mean. Tukey–Kramer multiple comparisons test.

**Table 2 ijms-25-09906-t002:** Antibodies and epitope retrieval used for immunolocalization.

Primary Antibodies	Dilution Immunohistochemistry (IH) Western Blot (WB)	Epitope Retrieval Buffer
Mouse anti-p21 (sc-817, Santa Cruz Biotechnology, Dallas, TX, USA)	1:25 (IH) 1:200 (WB)	1X Diva Decloaker (Biocare Medical, Pike Lane, CA, USA)
Biotin mouse anti-CD45 (103103, Biolegend, San Diego, CA, USA)	1:50 (IH) 1:200 (WB)	Tris buffer-EDTA pH 10
Mouse anti-PCNA [PC10] (GTX20029, GeneTex, Irvine, CA, USA)	1:300 (IH) 1:1000 (WB)	1X Diva Decloaker (Biocare Medical, Pike Lane, CA, USA)
Mouse anti-p-Histone H2A.X (sc-517348, Santa Cruz Biotechnology, Dallas, TX, USA)	1:25 (IH) 1:200 (WB)	1X Diva Decloaker (Biocare Medical, Pike Lane, CA, USA)
Goat anti-CYP11A1 (sc-18040, Santa Cruz Biotechnology, Dallas, TX, USA)	1:50 (IH) 1:500 (WB)	Tris buffer-EDTA pH 10
Goat anti-Actin (sc-1616, Santa Cruz Biotechnology Dallas, TX, USA)	1:5000 (WB)	
**Secondary antibodies**		
HRP horse anti-mouse (PI-2000, Vector Laboratories, Burlingame, CA, USA)	1:250 (IH) 1:1000 (WB)	
HRP rabbit anti-goat (R-21459, Thermo Fisher Scientific, Rockford, IL, USA)	1:250 (IH) 1:1000 (WB)	
Streptavidin, peroxidase (SA-5004, Vector Laboratories, Burlingame, CA, USA)	1:500 (IH) 1:1000 (WB)	

## Data Availability

Data are available upon request.

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
