# Peer review of "The Interstitial Gland as a Source of Pro- or Anti-Senescent Cells during Chinchilla Rabbit Ovarian Aging"

_ijms, 2024, doi:10.3390/ijms25189906_

Round 1

Reviewer 1 Report

Comments and Suggestions for Authors

I’m so happy to review the chance of the manuscript “The interstitial gland is a tissue rich in lipid droplets and lipophagy which may act as a source of pro- or anti-senescent cells, during ovarian aging”

In the study, the author reported the study of lipid droplets and lipophagy on the ovarian aging-like relationship between lipophagy implication in the interstitial gland.

Lipophagy is an important scientific topic and the specific role of ovary function. The analysis of Lipophagy could be a possibility for elucidation of ovarian aging. And author try to explain the lipidology related to ovarian aging.

Introduction

The author needs more clearly and logically explain the role of lipid droplets and lipophagy in ovarian aging. also more explain why interstitial gland the connectivity.

Each introduction part was not connected between sentence to sentence. Especially, line No 42 to 46 there are some evidences or not regarding lipophagy homeostasis. Then, the author introduces the connection point between lipophagy in the ovary and aging. Therefore, the author can more clearly explain, why need investigate lipophagy in ovarian aging.

The author shows several data. But most of the data is qualitative Results. Please add quantity data regarding lipid droplet or lipophagy-related gene or protein expression ratios by RT-PCR or western blot.

Figure 1 requests a re-oriented picture depending on the aging or on the level of aging (months).

Figures 2, and 3 present LD. But suggest to author add a numeric plot regarding LD size or number per ovary depending on the aging of the ovary. It’s more explains the relationship between LD and aging.

Figure 5. should be an additional experiment by western blot for analysis of protein expression ratios.

Table 1. It can be numerical data per tissue section. which is the author can be explained by number not “+” using ImageJ.

The author needs to compare data regarding lipophagy as you mention in the title. Lipofusion has the same properties as lipophagy or not. make right the word.

Figures 7, 8, and 9 also author can express the amount of analysis by western blot per tissue. If the author was not available western blot, the Author can calculate the positive cell number per section for numerical analysis.

Discussion

in this section was not parallel match the introduction and results section.

The author investigates the lipid droplet and lipophagy. But more focus on autophagy in this section. The author wants to have more discussion or autophagy. It should explain the relation point of lipophagy.

And request a final clear conclusion sentence more understandable for the reader.

Comments on the Quality of English Language

Moderate editing of English language required

Author Response

IJMS-3121864

Dear reviewers:

We are deeply grateful for the timely reviews provided to us. Your careful consideration of our work and insightful comments have been instrumental in our revisions. We have carefully reviewed the comments and implemented changes, seeking to carefully address each of them. The responses to the recommendations are given in a different color for each reviewer to facilitate the identification of the changes.

REVIEWER 1

Comments and Suggestions for Authors

I’m so happy to review the chance of the manuscript “The interstitial gland is a tissue rich in lipid droplets and lipophagy which may act as a source of pro- or anti-senescent cells, during ovarian aging”

In the study, the author reported the study of lipid droplets and lipophagy on the ovarian aging-like relationship between lipophagy implication in the interstitial gland.

Lipophagy is an important scientific topic and the specific role of ovary function. The analysis of Lipophagy could be a possibility for elucidation of ovarian aging. And author try to explain the lipidology related to ovarian aging.

Introduction

The author needs more clearly and logically explain the role of lipid droplets and lipophagy in ovarian aging. also more explain why interstitial gland the connectivity.

and

Each introduction part was not connected between sentence to sentence. Especially, line No 42 to 46 there are some evidences or not regarding lipophagy homeostasis. Then, the author introduces the connection point between lipophagy in the ovary and aging. Therefore, the author can more clearly explain, why need investigate lipophagy in ovarian aging.

Dear reviewer: Thank you for your suggestions; the modifications in the manuscript are tracked in red.

  1. We regret the lack of clarity; in the Introduction, we have clarified the importance of studying lipophagy in aging with special attention to the interstitial gland.
  2. We provide specific responses to the following three suggestions.

The author shows several data. But most of the data is qualitative Results. Please add quantity data regarding lipid droplet or lipophagy-related gene or protein expression ratios by RT-PCR or western blot.  and Figures 2, and 3 present LD. But suggest to author add a numeric plot regarding LD size or number per ovary depending on the aging of the ovary. It’s more explains the relationship between LD and aging, and The author needs to compare data regarding lipophagy as you mention in the title. Lipofusion has the same properties as lipophagy or not. make right the word.

R: We appreciate the suggestion to add lipophagy-related gene or protein expression dates. However, our main limitation in addressing this point is that our laboratory does not have access to antibodies or primers for real-time PCR. Besides this, standardizing the immunohistochemical test or quantitative real-time PCR and attaining results would take at least three more months. We are limited by the number of slides containing paraffin sections of the samples, as they have already been used in other immunohistochemistry studies that have not been published yet or may not be published at all. In addition, we do not have the material for RNA or protein extraction.

Your proposal to measure lipid droplets by age is fascinating. However, as mentioned in the methodology regarding the acquisition of semi-thin and thin sections, we used a fragment thickness of only 1.5 mm, which required the subdivision of the tissue during processing. The thin sections were too thick (100 nm), making it challenging to obtain the serial sections needed for an accurate count of the lipid droplets. While we could observe different sizes, the thickness prevented us from determining the three-dimensional levels of the lipid droplets. This could lead to the inaccurate assessment and quantification of the lipid droplets. We thus propose to defer the implementation of your valuable suggestions until the future characterization of lipophagy in the aging ovary.

Considering your observations, we have modified the title and removed the word lipophagy.

Figure 1 requests a re-oriented picture depending on the aging or on the level of aging (months).

R: In Figure 1 , we have included the ages in months for reference.

Figure 5. should be an additional experiment by western blot for analysis of protein expression ratios. We value your suggestion.

R: In Figure 5, it is shown that the interstitial gland foamy cells (IGFCs) did not test positive for CD-45, the pan-leukocyte marker, ruling out their classification as macrophages. However, they were positive for the enzyme CYP11A1, characteristic of steroidogenic cells. Previously, we reported the increased expression of CYP11A1 in the interstitial glands of mature rabbit ovaries. To validate the specificity of both antibodies, Western blot tests were conducted. Figure 10 includes appropriate densitometric or cell percentage quantifications.

Table 1. It can be numerical data per tissue section. which is the author can be explained by number not “+” using ImageJ.

R: For Table 1, we conducted a densitometric analysis for each tissue section using the ImageJ program.

Figures 7, 8, and 9 also author can express the amount of analysis by western blot per tissue. If the author was not available western blot, the Author can calculate the positive cell number per section for numerical analysis.

R: We validated the specificity of the antibodies with a Western blot. Unfortunately, we do not have the tissue for the RT-PCR or WB analysis of the reported samples. However, we can provide the appropriate densitometric or cell percentage analysis (Figure 10).

Discussion

in this section was not parallel match the introduction and results section.

The author investigates the lipid droplet and lipophagy. But more focus on autophagy in this section. The author wants to have more discussion or autophagy. It should explain the relation point of lipophagy.

R: We have modified the Discussion to better align with the Introduction and Results. In addition, we have included the relationship between autophagy and lipophagy in the Introduction and revisited it in the Discussion.

And request a final clear conclusion sentence more understandable for the reader.

R: We have modified the Conclusions to provide greater clarity.

Reviewer 2 Report

Comments and Suggestions for Authors

This research studied the interstitial gland in rodent ovaries and its involvement in ovarian senescence. The manuscript is interesting and useful in the attempt to understand ovarian aging at the cellular and subcellular level but there are some problems.

The abstract is unclear and rather difficult to understand. It needs to be reworked so that it can properly reflect the Results and Discussion.

The manuscript lacks a statement of the purpose or aim of the research. This should be added.

I have made further comments in the attached pdf.

Comments on the Quality of English Language

In the abstract the meaning of several sentences is obscured by language and grammar mistakes, which make its message very hard to understand. I have pointed out some of them in the attached pdf. The abstract should be corrected.

Author Response

IJMS-3121864

Dear reviewers:

We are deeply grateful for the timely reviews provided to us. Your careful consideration of our work and insightful comments have been instrumental in our revisions. We have carefully reviewed the comments and implemented changes, seeking to carefully address each of them. The responses to the recommendations are given in a different color for each reviewer to facilitate the identification of the changes.

REVIEWER 2

This research studied the interstitial gland in rodent ovaries and its involvement in ovarian senescence. The manuscript is interesting and useful in the attempt to understand ovarian aging at the cellular and subcellular level but there are some problems.

 R: Dear reviewer, we appreciate your insightful comments. The modifications in the manuscript have been highlighted in yellow.

The abstract is unclear and rather difficult to understand. It needs to be reworked so that it can properly reflect the Results and Discussion.

R: The Abstract has been reworded.

 The manuscript lacks a statement of the purpose or aim of the research. This should be added.

R: At the end of the Introduction, we have added the objective of the work.

I have made further comments in the attached pdf.

R:We have addressed the comments provided in the PDF; the corresponding revisions are highlighted in yellow in the revised version.

Comments on the Quality of English Language

In the abstract the meaning of several sentences is obscured by language and grammar mistakes, which make its message very hard to understand. I have pointed out some of them in the attached pdf. The abstract should be corrected.

R: We apologize for the inconsistency in the wording, which we have changed to facilitate improved comprehension. Moreover, the manuscript has already been reviewed and corrected by the English Editing department of MDPI.

Reviewer 3 Report

Comments and Suggestions for Authors

The subject of this manuscript (ijms-3121864) is interesting, as are the results obtained. However, this manuscript cannot be published in its current form and requires major revisions.

Reviewer’s suggestions:

1.       The title seems too long, please shorten it. My suggestion is: Interstitial gland as a source of pro- or anti-senescent cells during ovarian aging. If the authors want to emphasize the role of lipophagy in this process, the title can be changed to: Interstitial gland as a source of pro- or anti-senescent cells during ovarian aging - the role of lipophagy

2.       line 157. It is said: with atretic follicle  (atF), but the abbreviation should be AtF (as described in the photos and later in the manuscript).

3.       line 159 (Figure 2 captions), the description “as round and elongated tubular cristae mitochondria” should be removed – because these structures are not marked and are not visible in photo B. However, the description of photo C, in which mitochondria are visible and marked by the authors, should be left unchanged.

4.      Please check the scale bars in the Figure 3 captions, there are some errors there, for example: C) 2 µm and D) 200nm

5.       In line 200 (captions under Fig. 4) it is said: „Electron micrograph of interstitial gland foamy cells displays lysosome arm-like extensions", however, in the presented photos these structures are difficult to see, so they should be marked (for example with arrows). In addition, the scale bars in images D and E should be 500 and 200 nm, respectively.

6.       In lines 195-197, it is said: „Lipid-containing double-membrane vesicles, similar to autolipophagosomes, were apparent. However, no alterations were observed at the level of nuclear morphology (Fig. 4D-E). For the sake of clarity, please put the abbreviation (aphs) after the word autolipophagosomes, as this will immediately draw the reader's attention to the structures marked  in photos D and E. Moreover, there was a mistake in the description, because the proposed photos 4D-E do not show the morphology of the cell nucleus, photos 4A-B should be indicated.

7.      For what purpose did the authors use PAS staining (a staining method used to detect polysaccharides) in the ovaries? The authors do not refer to this staining at all in the rest of the manuscript. What was its purpose? Please explain

8.      line 215, authors should provide the name of the CYP11A1 enzyme and briefly characterize its role in steroidogenesis (in addition to the description given in the discussion)

9.       lines 236-237, it is said: „Qualitative analysis showed higher lipofuscin accumulation in the interstitial gland of mature rabbit ovaries, with OIGFC showing higher accumulation (Fig. 6, Table 1)”. This sentence should be changed to: “Qualitative analysis showed higher lipofuscin accumulation in the interstitial gland compared to other structures of mature rabbit ovaries.” Furthermore, the most intense SBB staining was observed in OIGFC cells (Fig. 6, Table 1). Moreover, currently, such subjective assessment of staining intensity (see Table 1) is not practiced and is not accepted by scientific journals. The authors should perform optical density analysis for all ovarian structures, as they have similarly already done for the interstitial gland (see Table 2). Please provide such data.

10.   The next sentence: “In addition, we performed quantitative analysis in the interstitial gland, showing a significant increase in accumulation from 18 months onwards, which continued at 24 and 36 months (p<0.01) (Table 2)”, is constructed incorrectly and may mislead the reader. This suggests that lipofuscin accumulation kept increasing with age (which is inconsistent with the data in Table 2, where it rather slightly decreases with age or remains constant). Of course, there is a significant increase compared with the 4-month-old group, but such a description is misleading. Please correct this description of the results.

11.  In my opinion, Ki-67 is a more specific marker for the study of cell proliferation than PCNA. In many publications PCNA is used, but it seems that this marker gives a higher percentage of dividing cells in the studies than Ki-67, many of these being in fact other processes. My experience confirms these observations, so I suggest using Ki-67 to detect cell proliferation in the future. PCNA is involved in many molecular processes, even apoptosis (especially in the ovaries).

12.  PCNA-positive interstitial cells should be marked with arrows in images E and H in Figure 7 because they are poorly visible.

13.  All stainings, including the intensity of immunohistochemical reactions for individual markers tested (PCNA, p21, H2AX, etc.), should be presented in quantitative form calculated based on optical density analysis. Especially if we compare them between given structures or the age of the animals studied. Visual observations should be presented and supported by quantitative analysis. This is of course laborious, but necessary for reliable presentation of results.

14.  line 370, should be written up to 18 months instead of 16

15.   lines 373-374, it is said: „The ovaries were quickly removed and fixed as follows: the left ovary was cut in half along a transverse plane and fixed in cold 4% paraformaldehyde (Sigma-Aldrich, St. Louis, MO, USA), overnight”. The authors should also add: then dehydrated, cleared in xylene, and embedded in paraffin.

16.   line 377, it is written: „Histological analysis was performed using PAS staining”. Why did the authors choose PAS for histological analysis?, furthermore there is no other additional information neither in the methodology nor in the description about the results regarding this reaction (apart from one published photo).

17.   According to manufacturer's datasheet, all antibodies indicated by the authors (except mouse anti-PCNA, Table 3) are not predicted to react with rabbit antigens, and are specific only for humans, mice and rats. So how can the authors be sure that this worked and gave a correct and specific reaction in rabbit tissues? Antibodies validation is needed.

Comments on the Quality of English Language

minor editing of English language is needed

Author Response

IJMS-3121864

Dear reviewers:

We are deeply grateful for the timely reviews provided to us. Your careful consideration of our work and insightful comments have been instrumental in our revisions. We have carefully reviewed the comments and implemented changes, seeking to carefully address each of them. The responses to the recommendations are given in a different color for each reviewer to facilitate the identification of the changes.

REVIEWER 3

Comments and Suggestions for Authors

The subject of this manuscript (ijms-3121864) is interesting, as are the results obtained. However, this manuscript cannot be published in its current form and requires major revisions.

Dear reviewer,

Your observations were not only helpful; they also significantly enriched our work. We deeply appreciate your contributions, and the time dedicated to reviewing our manuscript. The modifications in the manuscript are tracked in green.

Reviewer’s suggestions:

  1. The title seems too long, please shorten it. My suggestion is: Interstitial gland as a source of pro- or anti-senescent cells during ovarian aging.If the authors want to emphasize the role of lipophagy in this process, the title can be changed to: Interstitial gland as a source of pro- or anti-senescent cells during ovarian aging - the role of lipophagy

R: We appreciate your suggestion; we have changed the title of the manuscript according to the first of your suggestions.

  1. line 157. It is said: with atretic follicle(atF), but the abbreviation should be AtF (as described in the photos and later in the manuscript).

R: Line 165, changed (at F) to (At F).

  1. line 159 (Figure 2 captions), the description “as round and elongated tubular cristae mitochondria” should be removed – because these structures are not marked and are not visible in photo B. However, the description of photo C, in which mitochondria are visible and marked by the authors, should be left unchanged.

R: Thank you for your observation; we have deleted “as round and elongated tubular cristae mitochondria”.

  1. Please check the scale bars in the Figure 3 captions, there are some errors there, for example: C) 2 µm and D) 200nm

R: Line 189 has been corrected.

  1. In line 200 (captions under Fig. 4) it is said: Electron micrograph of interstitial gland foamy cells displays lysosome arm-like extensions", however, in the presented photos these structures are difficult to see, so they should be marked (for example with arrows). In addition, the scale bars in images D and E should be 500 and 200 nm, respectively.

R: We have added yellow arrows (fig. 4) and corrected the mistakes on line 209.

  1. In lines 195-197, it is said: „Lipid-containing double-membrane vesicles, similar to autolipophagosomes, were apparent”. However, no alterations were observed at the level of nuclear morphology (Fig. 4D-E). For the sake of clarity, please put the abbreviation (aphs) after the word autolipophagosomes, as this will immediately draw the reader's attention to the structures marked  in photos D and E. Moreover, there was a mistake in the description, because the proposed photos 4D-E do not show the morphology of the cell nucleus, photos 4A-B should be indicated.

R: Line 203 includes (aphs) after the word “autolipophagosomes”. Thank you for the correction on line 204.

  1. For what purpose did the authors use PAS staining (a staining method used to detect polysaccharides) in the ovaries? The authors do not refer to this staining at all in the rest of the manuscript. What was its purpose? Please explain

R: We appreciate your comment. We agree that the image of the PAS staining does not provide relevant information, so we have eliminated it along with its section in the Methodology.

  1. line 215, authors should provide the name of the CYP11A1 enzyme and briefly characterize its role in steroidogenesis (in addition to the description given in the discussion)

R: We have added the following lines 220-223: Cytochrome P450 family 11 subfamily A member 1 (CYP11A1), which catalyzes the first and rate-limiting step of the steroid hormones in the mitochondrial inner membrane, converting cholesterol into pregnenolone [24].

  1. lines 236-237, it is said: „Qualitative analysis showed higher lipofuscin accumulation in the interstitial gland of mature rabbit ovaries, with OIGFC showing higher accumulation (Fig. 6, Table 1)”.This sentence should be changed to: “Qualitative analysis showed higher lipofuscin accumulation in the interstitial gland compared to other structures of mature rabbit ovaries.” Furthermore, the most intense SBB staining was observed in OIGFC cells (Fig. 6, Table 1). Moreover, currently, such subjective assessment of staining intensity (see Table 1) is not practiced and is not accepted by scientific journals. The authors should perform optical density analysis for all ovarian structures, as they have similarly already done for the interstitial gland (see Table 2). Please provide such data.

R: We appreciate your comments and assistance. We have replaced the qualitative table with a quantitative table, named Table 1, and revised the presentation of the results.

  1. The next sentence: “In addition, we performed quantitative analysis in the interstitial gland, showing a significant increase in accumulation from 18 months onwards, which continued at 24 and 36 months (p<0.01) (Table 2)”,is constructed incorrectly and may mislead the reader. This suggests that lipofuscin accumulation kept increasing with age (which is inconsistent with the data in Table 2, where it rather slightly decreases with age or remains constant). Of course, there is a significant increase compared with the 4-month-old group, but such a description is misleading. Please correct this description of the results.

R: We have corrected the description of the results in the following lines: 242-245.

  1. In my opinion, Ki-67 is a more specific marker for the study of cell proliferation than PCNA. In many publications PCNA is used, but it seems that this marker gives a higher percentage of dividing cells in the studies than Ki-67, many of these being in fact other processes. My experience confirms these observations, so I suggest using Ki-67 to detect cell proliferation in the future. PCNA is involved in many molecular processes, even apoptosis (especially in the ovaries).

R: We highly appreciate your comment. One of our laboratory's top priorities will be to validate and standardize an anti-Ki67 antibody, enabling us to distinguish between proliferation and other cellular processes in the ovary.

  1. PCNA-positive interstitial cells should be marked with arrows in images E and H in Figure 7 because they are poorly visible.

R: We have added arrows to indicate positive cells in Figure 7.

  1. All stainings, including the intensity of immunohistochemical reactions for individual markers tested (PCNA, p21, H2AX, etc.), should be presented in quantitative form calculated based on optical density analysis. Especially if we compare them between given structures or the age of the animals studied. Visual observations should be presented and supported by quantitative analysis. This is of course laborious, but necessary for reliable presentation of results.RWe have provided the appropriate densitometric or cell percentage analysis (Figure 10).
  2. line 370, should be written up to 18 months instead of 16

R: Line 415 now says “18”.

  1. lines 373-374, it is said: „The ovaries were quickly removed and fixed as follows: the left ovary was cut in half along a transverse plane and fixed in cold 4% paraformaldehyde (Sigma-Aldrich, St. Louis, MO, USA), overnight”. The authors should also add: then dehydrated, cleared in xylene, and embedded in paraffin.

R: Line 420 now reads “then dehydrated, cleared in xylene, and embedded in paraffin”.

  1. line 377, it is written: „Histological analysis was performed using PAS staining”. Why did the authors choose PAS for histological analysis?, furthermore there is no other additional information neither in the methodology nor in the description about the results regarding this reaction (apart from one published photo).

R: We appreciate your comment. We consider that the image of the PAS staining does not provide relevant information, so we have eliminated it along with its section in the Methodology.

  1. According to manufacturer's datasheet, all antibodies indicated by the authors (except mouse anti-PCNA, Table 3) are not predicted to react with rabbit antigens, and are specific only for humans, mice and rats. So how can the authors be sure that this worked and gave a correct and specific reaction in rabbit tissues? Antibodies validation is needed.

 R: We performed antibody validation via Western blotting and have added the results in Supplementary Figure 1.

We apologize for the inconsistency in the wording, which we have changed to facilitate improved comprehension. Moreover, the manuscript has already been reviewed and corrected by the English Editing department of MDPI.

Round 2

Reviewer 2 Report

Comments and Suggestions for Authors

The paper can be accepted in the current form.

Author Response

Thank you for taking the time to review our manuscript.

Reviewer 3 Report

Comments and Suggestions for Authors

The authors improved the manuscript mostly in line with the reviewer's expectations, although I have a few comments:

It would be a good idea to indicate in the title what species of animal the research concerns.

Lines 272–273: It should be written: ........... in the comparison between a 4-month-old  and a 36-month-old rabbit (not 24-month-old as the authors indicated) and between ..... ......... (see Figure 10).

Line 274; please delete the repetition at the end of the sentence - Fig. 10B. See below:

(Fig. 10B, **p<0.01; Fig. 7L’, 7F’, and I‘; and Fig. 10B).

Line 290; it is said: .....and remained constant in all mature ages. However, such a statement is incomprehensible when analyzing the data from the graph (Fig. 10). I suggest deleting this entry.

Line 328; it should rather be SEM than ESM (?)

In the added Figure 10B, the authors should have included the unit (next to the OD of p21 designation on the Y-axis), which is usually an arbitrary unit.

Authors should include information on antibodies validation by Western blot in the methodology (in the subsection on immunohistochemistry) and cite the supplementary file.

Comments on the Quality of English Language

Minor editing of English language required.

Author Response

The authors improved the manuscript mostly in line with the reviewer's expectations, although I have a few comments:

 R: Thanks for your valuable comments. The modifications in the manuscript are tracked in green

It would be a good idea to indicate in the title what species of animal the research concerns.

R: We appreciate your suggestion; we changed the title of the manuscript

Lines 272–273: It should be written: ........... in the comparison between a 4-month-old  and a 36-month-old rabbit (not 24-month-old as the authors indicated) and between ..... ......... (see Figure 10).

R: Lines 272 and 273 were corrected

Line 274; please delete the repetition at the end of the sentence - Fig. 10B. See below:

(Fig. 10B, **p<0.01; Fig. 7L’, 7F’, and I‘; and Fig. 10B).

R: Line 274, we deleted the repetition at the end of the sentence - Fig. 10B

Line 290; it is said: .....and remained constant in all mature ages. However, such a statement is incomprehensible when analyzing the data from the graph (Fig. 10). I suggest deleting this entry.

R: We deleted this sentence

Line 328; it should rather be SEM than ESM (?)

R: Line 328, now: SEM

In the added Figure 10B, the authors should have included the unit (next to the OD of p21 designation on the Y-axis), which is usually an arbitrary unit.

R: In Figure 10B, we added arbitrary unit.

Authors should include information on antibodies validation by Western blot in the methodology (in the subsection on immunohistochemistry) and cite the supplementary file.

We added information on antibodies validation by Western blot in the subsection on immunohistochemistry (lines 478-493).

Thank you for taking the time to review our manuscript.